# TGV: Tabular Data-Guided Learning of Visual Cardiac Representations

## Abstract

Contrastive learning methods in computer vision typically rely on different views of the same image to form pairs. However, in medical imaging, we often seek to compare entire patients with different phenotypes rather than just multiple augmentations of one scan. We propose harnessing clinically relevant tabular data to identify distinct patient phenotypes and form more meaningful pairs in a contrastive learning framework. Our method uses tabular attributes to guide the training of visual representations, without requiring a joint embedding space. We demonstrate its strength using short-axis cardiac MR images and clinical attributes from the UK Biobank, where tabular data helps to more effectively distinguish between patient subgroups. Evaluation on downstream tasks, including fine-tuning and zero-shot prediction of cardiovascular artery diseases and cardiac phenotypes, shows that incorporating tabular data yields stronger visual representations than conventional methods that rely solely on image augmentations or combined image-tabular embeddings. Our results show that tabular-guided training produces strong unimodal image encoders, highlighting the potential of our approach for medical foundation model development.

## 1 Introduction

Biobanks provide large-scale multimodal medical datasets that can be leveraged to train medical foundation models. These datasets typically include imaging modalities, such as magnetic resonance (MR) or computed tomography (CT) scans, alongside structured tabular data describing demographics and clinical history. Despite the potential of image-tabular methods, integrating those two modalities remains limited, even though clinicians routinely combine such information for diagnosis. In cardiology, for instance, sex, age, and smoking status are key indicators of cardiovascular disease risk (7; 1), the leading cause of death worldwide (17). This highlights the importance of developing models that jointly leverage tabular and imaging data for improved clinical decision-making. However, extensive tabular information available in biobanks is often missing in practice due to time constraints in clinical workflows (6), motivating approaches that use tabular data only during training while enabling image-only inference. Contrastive learning has proven effective for multimodal data integration, particularly in image–text settings (19; 3). Extending this idea, Hager et al. (10) proposed using tabular data to supervise medical image encoders, but their formulation relied on rigid one-to-one sample pairing, overlooking clinical similarity between patients. Such strategies can introduce false negatives (13), where clinically similar patients are pushed apart in the embedding space. Recent findings further suggest that unimodal training can rival or surpass multimodal supervision in vision tasks (8; 14), motivating our vision-centric approach that exploits tabular data as training guidance rather than as a joint modality.

We introduce **T**ables **G**uide **V**ision (TGV), a contrastive learning framework that leverages tabular similarity to construct clinically meaningful image pairs for unimodal visual representation learning. Unlike prior multimodal approaches, TGV uses tabular data solely to guide pair selection during training, enabling unimodal prediction at inference. Furthermore, we propose a modified k-nearest neighbors (k-NN) aggregation method for zero-shot prediction, where class or phenotype estimates are obtained from the mean labels of the most similar reference embeddings. We evaluate TGV using cardiac MR images and tabular attributes from the UK Biobank (16), demonstrating strong performance on cardiovascular artery disease (CAD) classification and cardiac phenotype prediction.

## 2 Methodology

We follow the setting of SimCLR (4), using a vision encoder $E$ to obtain image embeddings $v \in \mathbb{R}^d$, followed by a projection head $f_v$ mapping them to $z \in \mathbb{R}^p$.

### 2.1 Defining Tabular-Guided Pairs

Each image $x_i$ in a batch of size $N$ is associated with tabular attributes $a_i = \{a_{con_i}, a_{cat_i}\}$. We compute a pairwise similarity matrix $S \in \mathbb{R}^{N \times N}$ by combining continuous and categorical similarities:

$$S = \lambda S_{con} + (1 - \lambda) S_{cat}, \tag{1}$$

where $S_{con}$ is derived from normalized Euclidean distances between continuous variables, $S_{cat}$ from cosine similarity of categorical attributes, and $\lambda$ balances their contributions.

### 2.2 Tabular Data–Guided Visual Learning

For each image $x_i$, the most similar samples within a threshold $h$ of the maximum similarity score in $S$ are defined as positives. The contrastive loss aligns representations of similar images while pushing dissimilar ones apart:

$$L = \frac{1}{N} \sum_{i=1}^{N} -\log \left( \frac{\sum_{j \in pos} \exp(\langle z_i, z_j \rangle / \tau)}{\sum_{j=1}^{N} \exp(\langle z_i, z_j \rangle / \tau)} \right), \tag{2}$$

where $\tau$ is a temperature parameter.

### 2.3 Zero-Shot Prediction

To enable zero-shot inference in a unimodal setting, we use a reference set $P = \{v_j\}$ of training embeddings with known tabular attributes. For an unseen image $x_i$, cosine similarity $s_{ij}$ is computed to all $v_j \in P$, and the target attribute is predicted as the mean value over the top-$K$ most similar samples:

$$\hat{a}_i = \frac{1}{K} \sum_{j \in \mathcal{N}_i} a_j, \tag{3}$$

where $\mathcal{N}_i$ denotes indices of the top-$K$ similar embeddings.

## 3 Experimental Setting & Results

### 3.1 Dataset

We train and evaluate our method on the UK Biobank population study (16), comprising 49,737 pairs of short-axis cardiac MR images and tabular data. The data are split into 39,975 training, 2,794 validation, and 6,968 test samples. Each MR volume includes 11 slices over 10 frames uniformly sampled from 50-frame cine sequences, zero-padded and cropped to $128 \times 128$. The tabular data contain 24 attributes (10 categorical, 14 continuous), including cardiac phenotypes such as left and right ventricular ejection fraction, and demographic and clinical information (e.g., sex, smoking status, and coronary artery disease (CAD) indicators). CAD attributes follow ICD-10 definitions from (10). For multi-label CAD prediction, we use a disease-balanced subset of 6,426 samples, considering only pre-scan diagnoses. Fine-tuning for cardiac phenotype prediction is performed on

| | CAD ↑ | | LVEF ↓ | | LVEDM ↓ | | LVEDV ↓ | | RVEF ↓ | | RVEDV ↓ | | MYOESV ↓ | |
|---|---|---|---|---|---|---|---|---|---|---|---|---|---|---|
| Model | ZS | FT | ZS | FT | ZS | FT | ZS | FT | ZS | FT | ZS | FT | ZS | FT |
| Mean-Guess | - | - | 4.81 | - | 17.85 | - | 29.54 | - | 4.73 | - | 26.73 | - | 17.85 | - |
| *Supervised* | | | | | | | | | | | | | | |
| ResNet50 (11) | - | 65.61 | - | 4.31 | - | 5.59 | - | 10.27 | - | 3.81 | - | 8.55 | - | 6.44 |
| *Image Augmentation* | | | | | | | | | | | | | | |
| SimCLR (4) | 62.05 | 71.68 | 4.72 | 3.49 | 13.01 | 5.25 | 23.26 | 9.82 | 4.71 | 3.18 | 21.89 | 8.25 | 13.48 | 6.06 |
| BYOL (9) | 56.99 | 67.32 | 4.92 | 3.99 | 16.43 | 5.74 | 27.81 | 10.31 | 4.95 | 3.39 | 26.29 | 7.99 | 16.19 | 5.96 |
| SimSiam (5) | 57.01 | 69.89 | 4.93 | 3.93 | 16.04 | 6.59 | 27.46 | 11.09 | 4.94 | 3.43 | 25.62 | 8.52 | 16.14 | 5.89 |
| Barlow Twins (18) | 55.12 | 65.01 | 4.90 | 3.57 | 16.54 | 6.12 | 27.17 | 11.01 | 4.97 | 3.39 | 25.98 | 8.35 | 16.52 | 6.14 |
| *Tabular Supervision* | | | | | | | | | | | | | | |
| MMCL (10) | 62,49 | 72.91 | 4.48 | 3.27 | 8.74 | 5.63 | 15.12 | 9.95 | 4.55 | 3.12 | 13.70 | 7.47 | 9.54 | 5.51 |
| *Tabular Guidance* | | | | | | | | | | | | | | |
| TGV (Ours) | **68.70** | **76.1** | **4.08** | **3.18** | **7.64** | **4.86** | **13.63** | **9.23** | **3.98** | **2.95** | **12.43** | **7.39** | **8.18** | **5.2** |

Table 1: Downstream task performance comparison for multi-label CAD classification evaluated using AUC and cardiac phenotype prediction (remaining columns, ↓) using MAE. ZS stands for zero-shot, FT for fine-tuning. The best result is shown in **bold**, while the second-best is underlined.

5,000 samples, with label quality checks following (2). A balanced reference set of 2,000 patients is used for zero-shot prediction, where predictions are averaged over the top 20% most similar embeddings for CAD classification and 2.5% for phenotype estimation, with percentages tuned on the validation set.

### 3.2 Tabular Guidance Outperforms Image Augmentation and Tabular Supervision

We benchmark our approach against both supervised and contrastive learning baselines. Specifically, we compare with a supervised ResNet50 (12) and five self-supervised methods: four image-only models, SimCLR (4), SimSiam (5), Barlow Twins (18), and BYOL (9), and one multimodal method, MMCL (10), which applies tabular supervision in a CLIP-like manner (15). All models are evaluated on two downstream tasks. The first is multilabel coronary artery disease (CAD) classification, evaluated using the area under the ROC curve (AUC), which provides a robust evaluation metric for imbalanced datasets with few pathological samples. The second task is cardiac phenotype prediction, evaluated using mean absolute error (MAE) across six attributes: left ventricular ejection fraction (LVEF), left ventricular end-diastolic mass (LVEDM), left ventricular end-diastolic volume (LVEDV), right ventricular ejection fraction (RVEF), right ventricular end-diastolic volume (RVEDV), and myocardial end-systolic volume (MYOESV). These metrics comprehensively assess model performance across both systolic and diastolic phases and capture all major cardiac regions. Table 1 summarizes results for zero-shot and fine-tuned evaluations, with zero-shot scores averaged across three representative patient sets $P$.

Our method consistently achieves the best results across all tasks and evaluation modes, demonstrating the effectiveness of integrating tabular information into visual representation learning. These findings indicate that forming pairs using clinically meaningful tabular attributes leads to stronger representations than conventional augmentation-based sampling. Furthermore, TGV surpasses tabular supervision, suggesting that enforcing a shared embedding space with tabular data can constrain the image encoder and reduce its capacity to extract informative visual cues. In contrast, tabular guidance encourages the model to internalize clinically relevant patterns directly from the images, resulting in a more expressive and semantically aligned visual encoder.

## 4   Conclusion

We present TGV, a contrastive learning paradigm leveraging tabular data to generate clinically meaningful pairs for training of visual representations. Our approach outperforms augmentations-based image only contrastive learning and tabular-supervision on CAD classification and cardiac phenotype prediction, highlighting the strength of our approach in a medical setting. Additionally, we propose a zero-shot prediction method compatible with unimodal image representations, overcoming a crucial limitation of those representations. TGV can be leveraged to train medical foundation models grounded on rich clinical information, paving the way for more robust and generalizable medical models.

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
