# OpenReview forum: "TGV: Tabular Data-Guided Learning of Visual Cardiac Representations"
_EurIPS.cc/2025/Workshop/MedEurIPS — EurIPS 2025 Workshop MedEurIPS Submission_

### Official Review · Reviewer_BmcK · 2025-10-29
**Review Comments**

**Rating:** 6
**Confidence:** 5

**Review:**

This paper presents a contrastive learning approach that uses tabular clinical data to guide the formation of image pairs during training. The method is evaluated on UK Biobank cardiac MRI data for CAD classification and cardiac phenotype prediction.

Somethings can be improved:
1. The paper mentions 24 tabular attributes but does not clearly specify which attributes are used in Equation (1) to compute the similarity matrix S. This is a critical design choice and represents the core contribution of the work. Are all 24 attributes used? Are certain clinically relevant attributes (e.g., age, sex, smoking status) weighted differently?

2. It would be helpful to see examples of which patients are grouped together based on tabular similarity to build intuition.

---

### Official Review · Reviewer_XqzM · 2025-10-31

**Rating:** 9
**Confidence:** 4

**Review:**

In my opinion, this is an excellent use case for a workshop abstract and would stimulate good discussion at the MedEurIPS workshop. Overall, the paper is clearly written and well motivated. The authors present a simple yet effective contrastive learning framework to train image representations using tabular data. Unlike standard CLIP-like approaches, which require a joint embedding space and may fail to capture similarity between patients, the proposed method uses tabular data similarities to construct meaningful image pairs and guide the training of visual encoders. This design is particularly relevant for clinical workflows, where tabular data are often available during training while image-only inference is needed at deployment.

The authors evaluate their framework on the UK Biobank population study and compare it with a supervised model and five self-supervised baselines, including a multimodal CLIP-like approach, across two clinical tasks. The preliminary results highlight the promise of the proposed framework for developing medical foundation models. It might be valuable to further explore how the choice of hyperparameters (lambda and h) influences tabular-guided visual learning, as well as how the size of the reference set affects zero-shot prediction and tuned cutoffs for selecting the most similar samples.

---

### Decision · Program_Chairs · 2025-10-31

**Decision:**

Accept (Oral)

**Comment:**

Both reviewers find the paper clear, well-motivated, and well aligned with the workshop theme. The proposed use of tabular data to guide contrastive learning in cardiac MRI is original and promising, with solid preliminary results.